# Comparing Moralities in the Abrahamic and Indic Religions Using Cognitive Science: Kindness, Peace, and Love versus Justice, Violence, and Hate

Aria Nakissa [1,2,3]

1   Department of Jewish, Islamic, and Middle Eastern Studies, Washington University in St. Louis,
    St. Louis, MO 63130, USA; arianakissa@gmail.com
2   Department of Anthropology, Washington University in St. Louis, St. Louis, MO 63130, USA
3   The Faculty of Social Sciences, Universitas Islam Internasional Indonesia, Jakarta 16416, Indonesia

**Abstract:** Recent cognitive science research indicates that humans possess numerous biologically rooted religious and moral intuitions. The present article draws on this research to compare forms of religious morality in the Abrahamic traditions (Judaism, Christianity, Islam) and the Indic traditions (Hinduism, Buddhism, Jainism). Special attention is given to moral teachings on kindness, peace, and love, as well as related teachings on justice, violence, and hate. The article considers how moral intuitions shape Abrahamic/Indic moral teachings, which, in turn, impact: (1) Abrahamic/Indic doctrines concerning politics, law, and war; (2) Abrahamic/Indic doctrines concerning individual ethics, and moral behavior proper to monastics and laypersons; and (3) Abrahamic/Indic doctrines concerning theological matters, such as the nature of the universe, souls, and deities.

**Keywords:** cognitive science; morality; violence; Judaism; Christianity; Islam; Hinduism; Buddhism; Jainism





## 1. Introduction

In modern Western societies, kindness, peace, and love are often seen as the essence of moral goodness. Meanwhile, violence and hatred are often seen as the essence of moral evil. This perspective has long shaped Western scholarship and popular discourse on religion. Since the Enlightenment, debates over whether religion is good or bad have focused on the question of whether religion encourages violence and hate (see, for example, Harris 2008; Cavanaugh 2009; Cronk 2009, esp. pp. 179–204; Atran 2010; Calhoun et al. 2011; Domínguez 2017). Such debates have inspired innumerable studies, which examine how various religions promote violence and hate—especially in the context of wars, ethnic conflicts, and systems of discriminatory legislation (e.g., Kuper 1990; Kolbaba 1998; Bartholomeusz 1999; Victoria 2006; Brekke 2006; Haynes 2009; Jerryson and Juergensmeyer 2010; Banchoff and Wuthnow 2011; Jenkins 2011; Murphy 2011; Corey and Charles 2012; Firestone 2012; Mayer 2013; Gier 2014; Jha 2016; Yu 2018; Afsaruddin 2022). Meanwhile, efforts to show that religion can be a force for good have given rise to studies which examine how various religions promote kindness, peace, and love (e.g., Jennings 1996; Keown 2005; Shah-Kazemi 2010; Raaflaub 2016; Cole 2018; Fiala 2018; Augustine and Wayne 2019). Modern Western writings often judge certain religions as morally superior to others based on the claim that they emphasize kindness, peace, and love over violence and hate. On these grounds, Buddhism, Jainism, and (sometimes) Christianity are judged as morally superior to Hinduism, Judaism, and Islam (see, for example, Harris 2008, pp. 7−13; Jerryson and Juergensmeyer 2010, esp. pp. 3–16; Kang 2014; Victoria 2018). Additionally, Western writings often suggest that modernity has had a positive moral effect on all religions—encouraging them to undergo liberalizing reforms which prioritize kindness, peace, and love over violence and hate (i.e., acceptance of secularism and human rights; see Taylor 1998; Mahmood 2006; Hurd 2012).

At the same time, it is typically recognized that violence, and perhaps even hate, can be morally good under certain conditions, namely, where they help promote justice. In both premodern and modern times, "justice" has widely been understood as follows. Justice entails giving people what they deserve, in terms of benefits and harms. This involves enforcing a suitable system of rights and duties (see, for example, Ricoeur 1995, pp. 29–30; Raphael 2001, esp. pp. 1–7; Olsaretti 2003; Beever 2004; Kristjansson 2006; Johnston 2011, esp. pp. 63–88; Wolterstorff 2015, esp. pp. 85–92). Thus, people deserve certain benefits in the form of rights. For example, an individual deserves to have his/her body protected from assault, an employee deserves his/her wages, and a child deserves parental care. People can also deserve certain harms—in the form of punishments—when they violate their duties. For example, a murderer can deserve to be killed, a cheating spouse can deserve to be divorced, and a traitor can deserve to be expelled. A just person gives people their deserved rights (e.g., gives employees their wages) and also gives people their deserved punishments for violating duties (e.g., punishes murderers for violating their duty not to murder). Justice has special significance in the domain of politics, where groups cooperate to inflict violence by imposing laws and fighting wars. Thus, it is typically held that laws should be designed in keeping with the principles of justice (Raphael 2001; Beever 2004; Rawls 1999; Johnston 2011). It is also typically held that wars should also be waged in keeping with the principles of justice—an ideal referred to as "just war" (Walzer 2006; Kelsay 2007, esp. pp. 97–124; Corey and Charles 2012; O'Driscoll 2015; Dwivedi 2017a).

It is widely recognized that the value of justice is linked to, but distinct from, values of kindness, peace, and love. However, scholars debate the precise relationship between these values (e.g., Ricoeur 1995; Mendus 1999; Nussbaum 2013; Gheaus 2017; Fedock 2021)—especially in the context of Christianity (Marcin 1984; Schoenfeld 1989; Woodhead 1992; Grant 1996; Jackson 2003; Wolterstorff 2015; Cochran and Calo 2017; also see Shapira 2018). One common view, with strong roots in Judaism and Christianity, is that the values of kindness, peace, and love encourage giving people something better than what they deserve. Put differently, these values encourage moral behavior, which goes beyond justice (see, for example, Ricoeur 1995; Raphael 2001, pp. 1–2; Wolterstorff 2015; VanDrunen 2017; Shapira 2018). For example, a stranger may not deserve to be sheltered in one's home, and an assailant might deserve violent punishment. Justice entails treating these people in accordance with deservingness ("just deserts"). However, commitments to kindness, peace, and love encourage one to go beyond justice by sheltering the stranger. Likewise, such commitments encourage forgiving the assailant, thereby sparing him/her from suffering the hate and violence s/he deserves. With this in mind, it is useful to distinguish between two clusters of morally significant concepts. On the one hand, there is kindness, peace, and love. On the other hand, there is justice, violence, and hate.

Cognitive science provides new insights into these two concept clusters and clarifies their place in various religious traditions. Cognitive science is the interdisciplinary scientific study of the human mind/brain. It draws on a range of fields including biology, psychology, neuroscience, cultural anthropology, archeology, and history. Cognitive science posits that the mind/brain contains a set of evolved mechanisms. These mechanisms produce general patterns of thought, emotion, and behavior which recur across different societies. Nevertheless, in each society, patterns are molded into more distinctive forms which reflect local culture, technology, and politics. "Intuition" is a key concept in cognitive science research. Intuitions are tendencies to adopt particular beliefs. They are largely unconscious, and are simply felt to be correct. Intuitions are biologically rooted in the sense that they are generated by evolved mechanisms. Since the 1990s, cognitive science research has generated two sizable literatures which deal with intuitions, and which are relevant to the study of comparative religion.

First, there is a cognitive science literature specifically on religion (i.e., the "Cognitive Science of Religion") (Guthrie 1993; Boyer 2001; Atran 2002; Barrett 2011; Bering 2011; McCauley 2011; Norenzayan 2013; Johnson 2016). This literature posits that evolved mechanisms produce a range of religious intuitions. These incline humans to believe in

particular religious phenomena, including spirit beings (Boyer 2001, pp. 137–67; Barrett 2004, pp. 31–60) and life after death (Bering 2011, pp. 111–30; Johnson 2016, pp. 121–22).

Second, there is a cognitive science literature on morality, which is closely related to the literature on religion (Joyce 2006; De Waal 2009; Baron-Cohen 2011; Haidt 2012; Greene 2014; McKay and Whitehouse 2015; Baumard 2016; Tomasello 2016; Curry et al. 2019). This literature posits that evolved mechanisms produce a range of moral intuitions. Such intuitions incline humans to believe that certain acts are morally good (e.g., honesty) or morally bad (e.g., theft) (Robinson et al. 2007; Parkinson et al. 2011; Hofmann et al. 2014; Boyer 2015).

Over the past two decades, scholars (often from the humanities) have begun to integrate cognitive science insights into nuanced studies of specific religious traditions. Hence, there are studies on Judaism (Newman 2018; Maiden 2020), Christianity (Malley 2004; Luhrmann 2012; Czachesz 2016), Islam (Atran 2010; Svensson 2014; Nakissa 2020a, 2020b), Buddhism (Pyysiainen 2009; Arnold 2012; Purzycki and Holland 2019), Hinduism (Goldberg 2007), and Confucianism (Reber and Slingerland 2011). More rarely, studies of this sort have employed a comparative approach. For instance, Teehan (2010) compares moral teachings in the Abrahamic religions, giving special attention to their teachings on violence.

The present article builds on existing scholarship, but also goes beyond it. The Abrahamic religions are a family of related traditions which includes Judaism, Christianity, and Islam. The Indic religions are a family of related traditions which includes Hinduism, Buddhism, and Jainism. These two families of religious traditions are arguably the most important in history. Moreover, today, over 75 percent of the world's population adheres to an Abrahamic or Indic religion (Pew Research Center 2012, p. 9). This article offers one of the first comparative studies of the Abrahamic and Indic traditions based on cognitive science insights. Comparing multiple traditions is beneficial in that it helps reveal larger patterns. This article argues that biologically rooted moral intuitions give rise to particular patterns in the Abrahamic and Indic traditions. At the most general level, these patterns are reflected in attitudes towards kindness, peace, and love versus justice, violence, and hate. At a more specific level, these patterns are reflected in: (1) doctrines concerning politics, law, and war; (2) doctrines concerning individual ethics, and moral behavior proper to monastics and laypersons; and (3) doctrines concerning theological matters, such as the nature of the universe, souls, and deities. To be sure, numerous existing studies address the preceding attitudes and doctrines. However, the studies do not explain these attitudes and doctrines in relationship to larger patterns produced by biologically rooted moral intuitions.

Before ending this introductory section, it is necessary to address a controversial theoretical issue, namely, the validity of generalization in the comparative study of religion. The Abrahamic and Indic religious traditions are vast heterogenous entities which have changed significantly over time.[1] It is impossible to generalize about these traditions without simplifying them in major ways. Moreover, since the 1970s, postcolonial scholarship has argued that simplistic generalizations have often been utilized to distort and denigrate non-Western religious traditions, while justifying the exploitation and political subjugation of non-Western peoples (e.g., Said 1978; Al-Azmeh 1993; Dirks 2001; Masuzawa 2005; Arjana 2020). Postcolonial ideas have had a particularly strong impact on the humanities. Thus, whereas earlier humanities-oriented scholarship on religion was more comfortable with large-scale generalizations, recent scholarship often takes the view that such generalizations are problematic or entirely invalid. By contrast, cognitive science research takes a far more positive stance on generalizations. This stance is supported by statistical data on global psychological variation (which is overlooked in most humanities-oriented scholarship).

Thus, within cognitive science and related disciplines, it is widely recognized that significant statistically measurable psychological variation exists between populations in different parts of the world (Nisbett 2004; Henrich et al. 2010; Heine 2016). Much of this variation pertains to religion and morality. Scholars have sought to quantitatively measure this variation using various methods. For example, survey questions on belief

and behavior can be used to measure how much a given population believes in God, hell, or angels (e.g., "Do you believe in God/hell/angels?", "What happens after people die?", "How often do you pray?", "How often to you go to church?"). Survey questions can also be used to measure how much a given population believes in moral obligations to care for kin or help the poor (e.g., "Do you feel morally obligated to care for grandparents/poor people?", "Do you live with your grandparents?", "How often do you give money to the homeless?"). Experiments are also useful. For instance, suppose one wishes to quantitively measure how strongly populations in two cities morally oppose stealing. This might be done by purposefully abandoning numerous wallets in each city and then calculating how frequently they are stolen (see, for example, Cohn et al. 2019). Results from surveys and experiments can be checked for accuracy against socioeconomic statistics (e.g., a group's expressed views about moral opposition to extramarital sex can be checked against the prevalence of sexually transmitted diseases among group members) (see, for example, Gray 2004; Obermeyer 2006; Becker 2019). Over the past two decades, researchers have produced a large number of quantitative studies on religion and morality across the globe (for studies on religion, see Saroglou 2010; Gervais and Najle 2015; Saucier et al. 2015; Schmitt and Fuller 2015; Inglehart 2018; Stagnaro et al. 2019; Henrich 2020; also see Shweder et al. 1997; Hofstede 2001; Inglehart and Welzel 2005; Schwartz 2006; for studies on morality see Oyserman et al. 2002; van de Vijver et al. 2006; Gelfand et al. 2011; Graham et al. 2011; Saucier et al. 2015; Inglehart 2018; Henrich 2020; Iurino and Saucier 2020; also see Shweder et al. 1997; Hofstede 2001; Inglehart and Welzel 2005; Schwartz 2006; Alesina and Giuliano 2014). Studies not only demonstrate that populations differ in significant statistically measurable ways regarding religion and morality. Studies also demonstrate that religious traditions are one major predictor—and arguably one major cause—of these differences (Inglehart and Baker 2000; Inglehart 2018; Rindermann 2018, pp. 323–67; Henrich 2020; Nakissa 2021).[2] Existing data supports the claim that different religious traditions (e.g., Christian, Hindu, Buddhist, Confucian) exert significant influence over contemporary populations, affecting them in distinctive statistically measurable ways (see, for example, Adamczyk and Hayes 2012; Inglehart 2018, pp. 36–59; Rindermann 2018). For example, even in secularized European countries, those with a Protestant heritage differ from those with a Catholic heritage. Hence, compared to Protestants, Catholics are more supportive of "traditional" values concerning religion and family (Inglehart and Baker 2000). Outside of Europe, past traditions of Protestantism and Catholicism likewise generate regular statistically measurable population differences (e.g., among Christians in Africa and India) (Henrich 2020, esp. pp. 3–17). Islam exerts a particularly significant influence on contemporary populations. In comparison to other religious groups, Muslims have higher levels of religious belief and place greater moral value on family bonds (see Inglehart and Baker 2000; Paldam 2009; Norris 2011; Nakissa 2021). Cognitive science scholarship holds that differences in doctrine are important for explaining the different effects that religious traditions have on societies (e.g., doctrines affirming an omnipotent God or the existence of hell) (see, for example, Shariff and Rhemtulla 2012; Norenzayan 2013; Henrich 2020).

To sum up, existing statistical data supports the view that, in many contexts, it is possible to make generalizations about major religious traditions. In other words, it is possible to speak about such traditions as distinctive entities, with distinctive doctrines, that produce distinctive effects on society. Based on this perspective, cognitive science research often sets forth generalizations about religious doctrines and their social effects.

Because this article is oriented by cognitive science, it sets forth some generalizations. In doing so, it cannot avoid some simplification of the religious traditions that it describes. There is no room to discuss all variant forms of the traditions. Instead, the article will concentrate on the most influential premodern forms of these traditions (e.g., Rabbinic Judaism, classical Sunni Islam).[3]

The remainder of this article is divided into five sections. In Section (I), I discuss cognitive science research on religious intuitions. I use these intuitions as framework

for introducing the basic theological doctrines of the Abrahamic and Indic traditions. In Section (II), I discuss cognitive science research on moral intuitions and explain how they shape attitudes towards kindness, peace, and love as well as justice, violence, and hate. In Section (III), I discuss the common view that justice-oriented morality is appropriate to the domain of politics, while kindness-oriented morality is appropriate to the domain of individual ethics. I also discuss how asceticism and monasticism relate to these domains. In Section (IV), I explain how biologically rooted intuitions shape Abrahamic religious views on morality, politics, individual ethics, monasticism, and theology. In Section (V), I explain how biologically rooted intuitions shape Indic religious views on morality, politics, individual ethics, monasticism, and theology.

## 2. Section (I): Cognitive Science, Religious Intuitions, and Abrahamic/Indic Theological Doctrines

Cognitive science research draws on several sources of evidence to establish that biologically rooted religious intuitions exist. Thus, psychological experiments indicate that religious intuitions emerge spontaneously in young children regardless of their upbringing and persist in adults. Cross-cultural surveys suggest that such intuitions recur in human societies across the globe. In some cases, religious intuitions can be linked to distinctive patterns of brain activity revealed by neuroimaging techniques. There is significant evidence for at least four religious intuitions. First, there is an intuition that "spirit beings" exist (i.e., beings which possess a mind, but lack an ordinary physical body) (Boyer 2001, pp. 137–67; Atran 2002, pp. 51–79; Barrett 2004, pp. 31–60). Second, there is an intuition that the soul is immortal, such that there is life after death (Bering 2011, pp. 111–30; Johnson 2016, pp. 121–22). Third, there is an intuition that all things in the universe were created purposefully by God (i.e., a supremely powerful spirit being) (Barrett 2004, pp. 75–93; Kelemen 2004; Petrovich 2019). Humans are inclined to believe that God is located above the Earth (Meier et al. 2007; Rihs et al. 2022) and that He created things specifically for the purpose of benefitting humanity (Preston and Shin 2021). Fourth, there is an intuition that doing a good deed will somehow result in a benefit for the doer (i.e., a reward). Similarly, doing a bad deed will somehow result in a harm for the doer (i.e., a punishment) (Callan et al. 2014; Johnson 2016, pp. 138–73).

These four intuitions can be thought of as "building blocks" (see McKay and Whitehouse 2015; Nakissa 2022). Different religious traditions incorporate some or all of these building blocks and mold them into a culturally specific form. This phenomenon is exemplified in the Abrahamic and Indic religious traditions. As noted above, it is impossible to describe these traditions without simplifying them. Nevertheless, some basic generalizations are helpful for our purposes, even if they are admittedly rough and imperfect.

Consider the Abrahamic traditions (see, for example, Peters 2004; Woodhead 2004; Silverstein et al. 2015; Stroumsa 2015; Goodman 2018; Nakissa 2019; Cohen 2020). These traditions affirm the existence of various lesser spirt beings, including angels (servants of God), demons, and (in some cases) spirits of the dead who wield power over the living (e.g., dead saints). The Abrahamic traditions assert that a single God created the universe out of nothing, and made all things in the universe with a purpose. While (Rabbinic) Judaism and Islam affirm a simple unified God, Christianity affirms a trinitarian conception of God. According to this conception, God simultaneously exists as one being and three different beings (i.e., the Father, the Son, and the Holy Spirit) in a manner which transcends ordinary human logic (Brower and Rea 2005; Bobrinskoy 2008; Emery and Levering 2011). The Abrahamic traditions accord a central place to prophets (e.g., Abraham, Moses, Muhammad) and prophet-like apostles (e.g., Peter, Paul, John). Just as angels are God's servants in heaven, prophets/apostles are God's servants on Earth. God wants humans to embrace specific theological and moral teachings. God reveals these teachings to prophets/apostles, who are tasked with communicating the teachings to the masses. In Christianity, it is further believed that God, incarnated as Jesus, temporarily descended to Earth to directly transmit these teachings Himself. In the Abrahamic traditions, God's

teachings are compiled and preserved in the form of scriptures (e.g., the *Hebrew Bible*, the *New Testament*, the *Quran*). The Abrahamic traditions assert that God will eventually bring an end to the world and then judge people based on their good and bad deeds (as well as their faith and character). God rewards good people by placing their immortal souls in heaven for eternity. He punishes bad people by placing their immortal souls in hell for eternity (although Jewish doctrine on the afterlife is more complex than Christian and Islamic doctrine).

We now come to the Indic religious traditions (see, for example, Babb 1996; Flood 1996; Dundas 2002; Gombrich 2006; Long 2009; Strong 2015). Compared to the Abrahamic traditions, the Indic traditions have been less strict in enforcing particular doctrines as "orthodoxy". Accordingly, Indic doctrines are less standardized and somewhat harder to summarize. Nevertheless, once again, some basic (if imperfect) generalizations are possible. Unlike the Abrahamic traditions, the Indic traditions are not averse to polytheism, and hence readily affirm many different types of spirit beings. These include gods of different ranks (e.g., Brahma, Vishnu, Indra, Yama), demons (e.g., Ravana, Mahishasura, Mara), nature spirits (e.g., Kubera, Gomukha), *nagas* (part-serpent part-human spirts), and ghosts or spirits of the dead. Hinduism, Buddhism, and Jainism share many of the same spirit beings, but give them somewhat different roles and classifications (see Appleton 2017). The Indic traditions affirm a doctrine of reincarnation, wherein an eternal soul (or something like it)[4] is repeatedly born in the form of different beings. Among these are spirit beings (e.g., gods, ghosts), humans of different ranks (e.g., kings, beggars), and animals (Chapple 2017). The beings are located in different realms, with humans and animals on Earth, gods in heavenly realms, and a class of unfortunate beings who dwell in hell realms. Reincarnation is governed by a law of *karma*. *Karma* dictates that an individual's good deeds automatically produce a reward and an individual's bad deeds automatically produce a punishment. Bad deeds cause an individual to be reborn in a lower form. For example, a human can be reborn as an insect or an inhabitant of hell. Good deeds cause an individual to be reborn in a higher form. For example, a human can be reborn as a god in heaven or as a higher-status human—such as a king or monastic. In this scheme, there is no eternal punishment, but there is a kind of eternal reward, which involves escaping the ongoing process of reincarnation (*samsara*), and entering into a state of permanent bliss (i.e., *mokhsha/nirvana*). The Indic traditions hold that the universe is eternal, but believe it is repeatedly created and destroyed in a cyclical manner. A dominant current within Hinduism asserts that a supreme spirit being (i.e., God) is responsible for this cyclical process of creation and destruction (e.g., Brahma, Vishnu, Shiva, the Goddess) (see Flood 2020). By contrast, Buddhism and Jainism generally reject the notion of a God (Analayo 2011; Harvey 2019; but see Obuse 2015). Instead, they posit that the universe is characterized by specific laws (e.g., the law of *karma*), and that these laws are responsible for the cyclical process of creation and destruction. Unlike the Abrahamic traditions, the Indic traditions do not accord a central place to prophets/apostles. Prophets/apostles are servants of God who are dependent upon God for their supernatural knowledge. In the Indic traditions, individuals can acquire supernatural knowledge without relying on God. This typically occurs through meditation. It is held that meditation produces extraordinary states of consciousness, which allow meditating individuals to access supernatural knowledge—including knowledge of particular scriptures (e.g., the *Vedas*, Mahayana sutras), knowledge of the universe's laws, and knowledge of past lives. Meditators pass this supernatural knowledge on to the masses. It is also believed that God (e.g., Krishna), gods, or god-like enlightened beings (*Buddhas*, *Tirthankaras/Jinas*) temporarily visit Earth and convey supernatural knowledge to human beings.

It will be noticed that the Abrahamic and Indic traditions mold general religious intuitions into distinctive religious doctrines. For example, while all traditions recognize spirit beings, these are of different types. The Abrahamic traditions hold that there are angels (i.e., servants of God), but no lesser gods, *nagas*, or god-like enlightened beings. By contrast, in the Indic religions, angels are peripheral or non-existent, while there are lesser

gods, *nagas*, and god-like enlightened beings. Similarly, while all traditions affirm life after death, the Abrahamic traditions hold that immoral souls are placed eternally in heaven or hell, while the Indic traditions hold that there is reincarnation (or permanent bliss upon escaping the process of reincarnation). Furthermore, the Abrahamic traditions hold that God is responsible for rewards and punishments, while the Indic traditions hold that such punishments automatically occur in keeping with the law of *karma*.

## 3. Section (II): Cognitive Science and Moral Intuitions

Cognitive science research draws on several sources of evidence to establish that biologically rooted moral intuitions exist. Thus, psychological experiments indicate that moral intuitions emerge spontaneously in young children regardless of their upbringing, and persist in adults (see, for example, Robinson et al. 2007; Rossano et al. 2011). Cross-cultural surveys suggest that such intuitions recur in human societies across the globe (see e.g., Haidt 2012; Inglehart 2018; Henrich 2020). In some cases, moral intuitions can be linked to distinctive patterns of brain activity revealed by neuroimaging techniques (see, for example, Parkinson et al. 2011; De Quervain et al. 2004). Studies further indicate that common forms of animal behavior (e.g., care for kin, incest avoidance) may underlie human intuitions (e.g., moral intuitions that one should care for kin and avoid incest) (see, for example, De Waal 2009; Nowak and Highfield 2011; Robinson et al. 2007).

The aforementioned cognitive science research provides new insights into morally significant concepts, such as kindness, peace, and love as well as justice, violence, and hate. Before proceeding further, these concepts should be defined with greater precision. Here, I offer definitions which are either found in the cognitive science literature, or which build on this literature. Kindness (i.e., "altruism", "prosociality") involves giving benefits to others (see De Waal 2009; Goetz et al. 2010; Haidt 2012, pp. 153–58; McKay and Whitehouse 2015, pp. 453–545; Curry et al. 2018). Understood broadly, giving benefits to others includes refraining from harming them, especially when they deserve it, or when it is in one's interests to do so. Thus, kindness includes refraining from punishing an assailant, and not stealing a neighbor's money even if one will not get caught (see Greene 2014, pp. 21–22). Refraining from harm (i.e., violence) is often described in terms of "peace" (or non-"aggression"; see Barratt and Felthous 2003; DeWall et al. 2011). Thus, it can be said that peace is an element of kindness.

Cognitive science research indicates that biologically rooted moral intuitions are tied to biologically rooted emotions. Here, kindness has received significant attention. Thus, research indicates that humans have an intuition that it is morally good to engage in kindness. This intuition is tied to emotions of love and compassion (De Waal 2009; Goetz et al. 2010; Baron-Cohen 2011; Haidt 2012, pp. 153–58; Crockford et al. 2014; McKay and Whitehouse 2015, pp. 453–54; Feldman and Bakermans-Kranenburg 2017; Decety 2021). With these facts in mind, it may be stated that kindness, peace, and love are linked together. Humans intuitively believe that it is morally good to engage in kind (and peaceful) behavior towards others. Love for others is an emotion which helps motivate/drive such behavior.

Violence (or "aggression") can be seen as the opposite of kindness (see Barratt and Felthous 2003; DeWall et al. 2011; Fiske and Rai 2015). Whereas kindness involves giving benefits to others, violence involves inflicting harms on others. Narrowly defined, violence entails inflicting harm/pain on others' bodies. Broadly defined, violence encompasses all acts which cause others harm/pain (e.g., stealing their property, slandering them). This article will use the broader definition.

Cognitive science research indicates that humans have intuitions that it is morally good to engage in violence under certain circumstances. Such violence is typically understood as punishment, which helps realize justice (Fehr and Gachter 2002; Baumard 2016; Haidt 2007, esp. p. 998; Fiske et al. 2007; Robinson et al. 2007; Haidt 2012, esp. pp. 150–79; Fiske and Rai 2015; McKay and Whitehouse 2015, esp. pp. 450–51; Cherry and Flanagan 2018, esp. pp. vii–xxxi; Sternberg and Sternberg 2008; also see Jordan et al. 2017; Fischer et al. 2018; Birondo 2022). Moreover, humans naturally experience emotions of hatred and anger,

which drive them to mete out violent punishment when appropriate (see Fehr and Gachter 2002; De Quervain et al. 2004; Robinson et al. 2007; Du and Chang 2015).

Cognitive science holds that cooperation is the key to understanding human morality. By extension, it is the key to understanding morally significant concepts, such as kindness, peace, love, justice, violence, and hate (see Nowak and Highfield 2011; Rai and Fiske 2011; Greene 2014, pp. 22–23; Fiske and Rai 2015; Curry et al. 2019). Human survival requires basic material goods (i.e., property) to nourish the body (i.e., food, drink) and protect it from the elements (e.g., shelter, clothing). Historically, humans have had to cooperate to produce these material goods. They have also had to cooperate to protect these goods and their bodies from attacks by others (e.g., animal predators, rival human groups). Cognitive science posits that, because cooperation enhances fitness, humans evolved tendencies to form and preserve particular types of cooperative social relationships. These include relationships between parents and children (Crockford et al. 2014; Feldman and Bakermans-Kranenburg 2017); relationships between blood kin more generally (tied to kin selection; Nowak and Highfield 2011, pp. 95–112; Curry et al. 2019); relationships between mates/spouses (tied to pair bonding; Feldman 2012; Buss 2016; Schacht and Kramer 2019); relationships between friends (with a history of cooperation; Brown and Brown 2006, pp. 13–14; Feldman 2012); and relationships between members of a group defined by shared characteristics (e.g., shared culture) (Haidt 2012, pp. 161–64; Tomasello 2016, pp. 85–134; Clark et al. 2019).

Cognitive science research holds that humans evolved various moral intuitions and emotions to help preserve cooperative social relationships (Nowak and Highfield 2011; Rai and Fiske 2011; Greene 2014, pp. 22–23; Fiske and Rai 2015; Curry et al. 2019). Thus, humans evolved a moral intuition that it is good to treat others kindly. Relatedly, they evolved emotions of love for others. Nevertheless, researchers also note that human kindness is of different types (see Curry et al. 2018). "Parochial kindness" is kindness directed towards specific individuals, including kin, mates, friends, and group members. "Universal kindness" is kindness directed towards all. Universal kindness is kindness in its most absolute and unlimited form. Research indicates that, owing to one or more moral intuitions, humans believe that it is good to engage in both parochial and universal kindness (De Waal 2009; Goetz et al. 2010; Baron-Cohen 2011; Haidt 2012, pp. 153–58; Crockford et al. 2014; McKay and Whitehouse 2015, pp. 453–54; Feldman and Bakermans-Kranenburg 2017; Decety 2021). However, humans have stronger inclinations towards parochial kindness. Hence, they tend to believe that there are special moral duties to kin, mates, friends, and group members. Humans likewise feel special love for these persons (see, for example, Feldman 2012; Clark et al. 2019; Curry et al. 2019). Accordingly, humans make unique efforts to preserve cooperative social relationships with these persons. For example, siblings (or friends) feel a special love for one another, and hold that they have a special moral duty to treat each other kindly. This motivates them to cooperate (e.g., in producing food, defending family land). Unlike parochial kindness, universal kindness requires giving benefits to strangers/outsiders, and thereby going beyond one's existing relationships. Doing this provides a way of forming new relationships—of acquiring new mates, friends, group members, and so on.

The distinction between parochial and universal kindness is important for understanding justice. In keeping with parochial kindness, humans acknowledge special moral duties of kindness towards kin, mates, friends and group members. In other words, these individuals deserve some significant measure of kindness, and associated benefits, as a right. For example, a child has a right to care, food, and shelter from his/her parents. A group member has a right to aid and protection from other group members. Matters are different for strangers. Although strangers/outsiders deserve some kindness, they deserve less kindness, and hence, fewer rights. Consequently, whereas justice requires showing significant kindness to kin, friends, and the like, it does not require showing such kindness to strangers/outsiders.

Although kindness and love play an essential role in preserving cooperative social relationships—taken alone—they are often not sufficient for this purpose. Certain behaviors threaten all such relationships, and are especially common beyond the circle of close kin. Thus, cooperative social relationships can be undermined by cheating, wherein individuals take benefits from others, but do not provide benefits in return (i.e., lack of reciprocity). Relationships can also be undermined through aggression, wherein individuals physically attack others and take their material goods (i.e., property) (see Nowak and Highfield 2011, pp. 21–112; Greene 2014, pp. 21–22; Curry et al. 2019). Cheating and aggression harm others and deny them the resources they need to survive, causing them to die or flee, with the result that they are no longer available to participate in cooperative endeavors.

Cognitive science holds that humans evolved moral intuitions and emotions, which prevent behavior that undermines or destroys cooperative relationships. Thus, there are moral intuitions which cause humans to believe: (1) that it is morally bad to engage in behaviors which undermine or destroy cooperation; (2) that it is morally good to punish individuals who engage in such morally bad behaviors. Punishment entails violence—inflicting different types of harms on the morally bad (e.g., attacking their bodies, confiscating their property, destroying their reputations, shunning them).

For instance, there is strong evidence that humans have moral intuitions concerning physical assault (Robinson et al. 2007; Baumard 2016, pp. 73–74); theft of property (Robinson et al. 2007; Rossano et al. 2011; Boyer 2015); and cheating (Fehr and Gachter 2002; Haidt 2007, p. 998; Robinson et al. 2007; Nowak and Highfield 2011, pp. 21–50; Curry et al. 2019, p. 107). All of these acts are regarded as morally bad, such that it is morally good to punish those guilty of them. Moreover, humans naturally feel emotions of anger and hatred towards those who engage in the preceding behaviors, and experience pleasure when the behaviors are punished. These emotions drive individuals to carry out punishments even if doing so comes at a personal cost (see Fehr and Gachter 2002; De Quervain et al. 2004; Robinson et al. 2007; Du and Chang 2015). Notably, humans naturally punish in keeping with a principle of proportionality. Hence, they regard some acts as more severe in their moral badness than others. More severe wrongs (e.g., murder) deserve harsher punishment, while less severe wrongs (e.g., theft) deserve lighter punishment (Robinson et al. 2007, esp. pp. 1636–37; Baumard 2016, pp. 73–74).

Although some cooperative social relationships are found between a small number of individuals (e.g., two friends, five members of a family), others involve a larger group. Put differently, members of a group frequently cooperate together. Accordingly, some of the most important moral intuitions concern groups. Thus, humans naturally and unconsciously divide the world into groups defined by shared characteristics (e.g., shared blood, cultural practices, religious practices). Humans also assume that such groups are competing with one another for power and resources. As indicated above, humans have an intuition that it is morally good to show kindness to one's group. This involves protecting the group, its members, and its distinctive way of life (e.g., shared cultural/religious practices). It also involves helping the group compete against other groups for power and resources (Haidt 2012, pp. 161–64; Tomasello 2016, pp. 85–134; Clark et al. 2019; also see Huang and Han 2014). The phenomenon at issue is often referred to as "tribalism," "patriotism," or "nationalism". Humans consider behavior which threatens one's group, its way of life, and its interests to be morally bad, such that it is morally good to punish those guilty of such behavior. When threats come from other groups, it is morally good to punish and subjugate them through war and/or discriminatory laws (e.g., laws designed to restrict their power, wealth, and influence). Threats can also come from individuals within one's own group, who ally with members of other groups, or who reject one's group's shared practices, thereby undermining its way of life. It is morally good to punish these "traitors". Emotions of hate and anger help motivate individuals to punish (often violently) other groups and traitors within one's own group (see Haidt 2012, pp. 161–64; Tomasello 2016, pp. 85–134). Such violence is typically seen as consistent with, if not required by, justice

(e.g., as in notions of "just war"; see Walzer 2006; Kelsay 2007, esp. pp. 97–124; Corey and Charles 2012; O'Driscoll 2015; Dwivedi 2017a).

## 4. Section (III): Differing Moral Codes in the Context of Politics, Individual Ethics, and Asceticism

As explained above, kindness, peace, and love play an essential role in preserving the cooperative social relationships necessary for human survival. The same is true of justice, violence, and hate. Nevertheless, in many cases, these two sets of concepts can be in tension or conflict.

Recall that justice entails giving people what they deserve. Moreover, a key feature of justice is limits. Thus, there are limits to how many benefits and harms an individual deserves. In other words, his/her rights are not infinite, and his/her punishments should be proportionate. Meanwhile, the values of kindness, peace, and love encourage behavior that goes beyond justice—giving people something better than what they deserve (see, for example, Ricoeur 1995; Raphael 2001, pp. 1–2; Wolterstorff 2015; VanDrunen 2017; Shapira 2018). Such behavior typically takes the form of charity or forgiveness. Charity entails giving others more benefits than they deserve (e.g., feeding and sheltering strangers). Forgiveness entails giving others fewer harms than they deserve (e.g., foregoing punishment of assailants). Significantly, the values of kindness, peace, and love have no obvious limits because they are not constrained by deservingness/justice. Thus, an individual can give varying amounts of charity—up to extreme quantities. A woman might donate some of her wealth or all of her wealth; she might donate some of her time (e.g., at a soup kitchen) or all of her time; she might donate one of her bodily organs, or all of them. Similarly, an individual can give varying amounts of forgiveness—up to extreme quantities. A man might forgive and refuse to punish someone who has insulted him, or not paid back a loan, or punched him, or killed his family. Because the values of kindness, peace, and love lack limits, a given individual must exercise some measure of free personal choice in determining how much kindness (e.g., charity, forgiveness) to perform. The term "supererogatory" is commonly used to describe morally good actions which go beyond duties (determined by justice), and which are freely chosen (see Urmson 1958; Kawall 2009).

Throughout history, many thinkers have held that in moral matters it is necessary to differentiate a domain of politics/government from a domain of individual ethics (see Ricoeur 1995; Raphael 2001, pp. 1–2; Wolterstorff 2015; VanDrunen 2017; Shapira 2018). The domain of politics/government encompasses law and war, and gives special priority to justice (see Rawls 1999; Raphael 2001; Beever 2004; Walzer 2006; Johnston 2011; Corey and Charles 2012). It also explicitly or implicitly endorses significant forms of violence and hate, which are essential aspects of law and war. Recall that violence and hate are essential for preserving cooperative social relationships. Among those who deserve to suffer violence and hate are parents who do not fulfill their children's rights, husbands who do not fulfill their wives' rights, and group members who do not fulfill the rights of other group members (e.g., to aid, protection). Similarly, among those who deserve violence and hate are friends who harm friends by stealing, and group members who harm other group members by assault. The law ensures that those who commit such offenses get what they deserve. The law makes them suffer violence through punishment. The law also actively fosters public hatred against them by exposing their crimes, shaming them, stigmatizing them, and formally declaring them to be "criminals", "felons", and "evildoers" (see Braithwaite 1989; Book 1999). As noted earlier, humans naturally feel hatred and anger towards those they know have committed morally bad actions, such as cheating, theft, and assault.

Because the domain of politics centers on justice, it is very concerned with limits based on deservingness. These limits are specified through the mechanism of law. Law clarifies precisely which rights and punishments a person deserves (e.g., spousal maintenance of 100 dollars per month, a punishment of 50 lashes). In doing so, law draws on universal

human moral intuitions (often identified with "natural law"), but adapts them to a specific social/cultural context.

Although the domain of politics does not exclude kindness, peace, and love, these are to be restricted. For were things otherwise, law and war would be impossible, with destructive results. For example, suppose a government resolved to kindly forgive all crimes. This would encourage citizens to engage in cheating, theft, and assault. Cooperative relationships would weaken, and citizens would destroy one another. Likewise, suppose a government resolved upon disarming itself and foregoing all violence (including self-defense) out of a commitment to peace and love. This would encourage devasting armed invasions from foreign groups. Any sustainable society requires significant violence and hatred directed towards moral evildoers (e.g., criminals) within and foreign enemies without.

However, as long as the government maintains justice through law and war, it will not destroy a society if certain individuals outside the government place much less emphasis on justice. Here, it is possible to speak of a distinctive domain of individual ethics. Although justice is present within this domain, it is not necessarily the dominant value. Rather, individuals may prioritize kindness, peace, and love. Such individuals, thereby, go beyond justice. However, given the absence of clear limits, each individual must voluntarily choose exactly how much kindness (e.g., charity/forgiveness) s/he wishes to perform. Each individual's actions may, thereby, be described as "supererogatory" (see Urmson 1958; Kawall 2009).

To sum up, it is useful to distinguish between two moral codes. A "justice-oriented" moral code gives comparatively greater emphasis to justice without excluding kindness (peace and love). A "kindness-oriented" moral code gives comparatively greater emphasis to kindness (peace and love) without excluding justice. The first code characterizes the domain of politics, while the second code characterizes the domain of individual ethics. (Interestingly, some cognitive science research also suggests that the justice-oriented moral code is more central to "conservative" social/political ideologies, while the kindness-oriented moral code is more central to "liberal" social/political ideologies; see Haidt 2012; Lakoff 2016).

Although the domains of politics and individual ethics are distinctive, there is no need to assume that the boundaries between them are clear. Moreover, there is no need to assume that the two domains cannot overlap or coexist with one another. In fact, they typically do overlap and coexist, albeit in ways that can engender tension and conflict.

To appreciate how the two domains (and their moral codes) coexist, attention must be given to the subjects of asceticism and monasticism. Here, I make some preliminary claims which relate to patterns found within the Abrahamic and Indic traditions. Evidence for these claims will be presented in the following two sections, which are devoted to Abrahamic and Indic religious doctrines.

Asceticism entails pursuing otherworldly salvation at the expense of worldly affairs. Because humans are a biological species, their conception of worldly affairs is closely tied to survival and reproduction. All Abrahamic and Indic religions value otherworldly salvation, and hence embrace some measure of asceticism.

Asceticism can take forms of greater and lesser intensity. More intense forms of asceticism may be described as "monasticism". Practitioners of monasticism are "monastics". Male and female monastics are often referred to (respectively) as "monks" and "nuns". In monasticism, there is overwhelming concern with otherworldly salvation, and minimal concern with worldly affairs, such as survival and reproduction. Thus, monastics frequently renounce sex (needed for reproduction). They also pay little attention to their bodily health (linked to survival). Accordingly, they often expose their bodies to hardships such as fasting, sleep deprivation, and self-mutilation. They also take vows of poverty, thereby depriving themselves of property needed for survival—such as food, shelter, and adequate clothing (i.e., they favor shabby clothing or nakedness). Full vows of poverty oblige monastics to roam around homeless, with their bodies exposed to the elements,

begging for food (see Karamustafa 1994; Wimbush and Valantasis 1998; Innemee 2005; Olson 2008).

Because monastics are minimally concerned with the worldly affairs of survival and reproduction, they are also minimally concerned with the cooperative social relationships needed to ensure survival and reproduction. Consequently, monastics frequently renounce mating/marriage relationships and kin relationships (e.g., they abandon their families). They also often abandon responsibilities for protecting relationships and society through politics, law, and war. Rather, they distance themselves from the domain of politics and focus on the domain of individual ethics.

Generally speaking, individuals are not forced to be monastics. Rather, it is something they choose. Part of this choice typically involves voluntarily committing themselves to an enhanced kindness-oriented moral code (i.e., they must be extra kind). Why do monastics embrace such a code? There are likely two complementary reasons. First, they believe that otherworldly salvation comes as a reward for morally good behavior (i.e., God or *karma* will reward such behavior). Moreover, given that kindness goes beyond justice, monastics hold that the kindness-oriented code prescribes behavior which is morally better than what is prescribed by the justice-oriented code. Second, monastics are unconcerned with worldly affairs, such as their own survival and reproduction. For instance, not caring about his/her property, a monastic can give it all away to others as charity. The monastic can also allow others to steal his/her property and forgive them. Indeed, the monastic can allow others to assault or kill his/her body and forgive them. In taking such a stance, the monastic sets aside concerns with justice. S/he does not demand the rights that s/he deserves, and does not demand the punishments that others deserve.

The fact that monastics do not embrace the justice-oriented moral code, or partake in the domain of politics, does not necessarily mean that they consider these things to be morally bad. Rather than conceptualizing morality in terms of good versus bad, it is also possible to think in terms of superior good versus inferior good. Monastics can hold the view that they are concerned with the superior goal of otherworldly salvation, and that this is best achieved through the kindness-oriented moral code, which is morally superior to the justice-oriented code. Meanwhile, laypersons are concerned with the inferior goal of worldly affairs, which necessitates partaking in the domain of politics and embracing its inferior justice-oriented moral code. According to this perspective, laypersons and their code are morally inferior to monastics and their code. Yet, laypersons and their moral code are not necessarily morally bad. They may even be morally good. Such a perspective operates to legitimate a society where laypersons and monastics coexist or cooperate—each abiding by a code that is morally good though unequal in value.

## 5. Section (IV): Patterns in the Abrahamic Religious Traditions

Biologically rooted moral intuitions give rise to particular patterns in the Abrahamic religious traditions. At the most general level, these patterns are reflected in attitudes towards kindness, peace, and love versus justice, violence, and hate. At a more specific level, these patterns are reflected in: (1) doctrines concerning politics, law, and war; (2) doctrines concerning individual ethics, and moral behavior proper to monastics and laypersons; and (3) doctrines concerning theological matters, such as the nature of the universe, souls, and deities. This section will explore the preceding patterns.

The Abrahamic traditions value a kindness-oriented moral code defined by kindness, peace, and love. They likewise value a justice-oriented moral code, which legitimates violence and hatred under certain circumstances. However, there are differences in emphasis. Here, it is useful to distinguish between (Rabbinic) Judaism and Islam on the one hand, and Christianity on the other. Judaism and Islam place comparatively greater emphasis on the justice-oriented moral code whereas Christianity places comparatively greater emphasis on the kindness-oriented moral code. These differences in emphasis shape doctrines on politics, individual ethics, monasticism, and theology. Let us examine these issues in more depth, beginning with a discussion of Judaism and Islam.

Judaism and Islam accept the domain of politics as perfectly legitimate, and hold that proper laws and wars are morally good. The most important figures in Judaism are the prophets Moses and David. Moses is a divinely inspired law-giver, who also served as a political-military leader. David is a warrior king. Islam's most important figure is Muhammad, who is understood to be like Moses. Hence, he is a divinely inspired law-giver, who also served as a political-military leader.

In Judaism and Islam, religiosity centers on adherence to divinely inspired laws given to prophets (Berger 1998; Neusner and Sonn 1999; Hallaq 2009; Licari 2019; Nakissa 2019). The Jewish law (given to Moses) is the "Torah", while the Islamic law (given to Muhammad) is the "Sharia". As noted earlier, law is primarily concerned with justice. The fact that Judaism and Islam center on law reflects their strong emphasis on justice. It may also be said that Judaism and Islam are deeply concerned with preserving social relationships. Thus, Jewish and Islamic laws are mechanisms for preserving parent-child relationships, marriage relationships, and kin relationships. These laws also preserve Jews and Muslims as groups united by a distinctive way of life.

Jews and Muslims believe their laws embody justice. This justice is based on a set of core moral teachings, which are associated with particular texts and doctrines. Core moral teachings in Jewish law are associated with the Ten Commandments (Coogan 2014) and the Seven Laws of Noah (Rosenberg 2003; Novak 2011). The Ten Commandments and the Noahide Laws prescribe worship/reverence of one God—prohibiting idolatry and disrespect for God's name. They also prohibit killing, stealing, sexual immorality (especially adultery), and spreading damaging lies about others. Core moral teachings in Islamic law are associated with the medieval theory of *Maqasid al-Sharia* (i.e., the theory of the Sharia's overriding aims). According to this theory, Islamic law has six[5] overriding aims, namely, (1) preservation of the monotheistic Islamic religion; (2) preservation of human life and bodily integrity (through the prohibition of murder and assault); (3) preservation of family lineage (through the prohibition of adultery and sexual immorality); (4) preservation of mental functions (through the prohibition of intoxicant consumption); (5) preservation of property (through the prohibition of theft); and (6) preservation of reputation (through the prohibition of slander) (Hallaq 1997, pp. 112–13, 162–87; Opwis 2010). Note that there is much overlap in the moral teachings of Jewish law and Islamic law, and also that many teachings can be linked directly or indirectly to biologically rooted moral intuitions.

Jewish law and Islamic law prohibit various forms of aggression/violence, such as murder, theft, and the like. These actions are regarded as morally bad in general; although they are particularly bad when directed at members of one's own religious group. Judaism and Islam hold that it is praiseworthy to feel hatred and anger towards those who engage in bad actions (e.g., *Psalm* 139:21, *Proverbs* 8:13; *Quran* 60:4, 48:29). Such actions merit punishment and shaming under the law. Hatred and anger help ensure proper punishment is implemented.

Jewish law and Islamic law endorse notions of "just war" (i.e., war constrained by moral principles of justice). Both laws endorse defensive war and also certain kinds of offensive war (Kelsay 2007, pp. 97–124; Jenkins 2011; Hallaq 2009, pp. 324–41; Firestone 2012; Walzer 2012; Afsaruddin 2022). During war, it is permissible to use violence against other groups; although such violence is to be limited. This is because even members of foreign groups have rights and do not deserve to be harmed in a careless manner. For example, there are restrictions on killing women, children, and the elderly (Solomon 2006; Kelsay 2007, pp. 97–124; Hallaq 2009, pp. 327–31).

Significantly, Jewish law and Islamic law prescribe many ritual/symbolic rules. For example, there are rules related to dress (e.g., beards, veils), bodily markings (e.g., circumcision), diet (e.g., kosher foods, halal foods), and purity/hygiene (e.g., ablution, bathing, shaving). These rules function to mark off Jews and Muslims as unique groups held together by a distinctive shared set of religious/cultural practices.

Although Jewish law and Islamic law center on justice, they also encourage forms of kindness which go beyond justice—especially in the domain of individual ethics. More

specifically, both laws require treating all people with justice, but also exhort believers to voluntarily perform supererogatory acts of kindness (e.g., charity, forgiveness) (Zaroug 1985; Shapira 2018; Nakissa 2019, p. 99).[6] At the same time, both Judaism and Islam make strong distinctions between groups. Supererogatory acts of kindness aiding members of one's own religious group are strongly and unequivocally encouraged. Members of foreign groups are owed justice, and there is some encouragement to aid them with supererogatory acts of kindness. Yet such encouragement is qualified and limited.

Judaism and Islam generally lack the more intense forms of asceticism associated with monasticism. This is because Jewish law and Islamic law strongly encourage non-monastic behaviors related to survival and reproduction (e.g., marriage, property ownership, the waging of wars). Admittedly, (Rabbinic) Judaism and Islam birthed ascetic movements focused on mysticism (Kabbalah, Sufism) (see Trimingham 1971; Schimmel 1975; Greenspahn 2011; Nakissa 2019). Nevertheless, these religions did not develop a distinct monastic class with a separate moral code (e.g., a code mandating enhanced kindness).

The Jewish and Islamic emphasis on justice is reflected in their theologies. These religions conceptualize God as a king who lays down a law and then judges people with justice according to their deeds. God rewards those who obey the law by fulfilling their duties. He punishes those who disobey the law by violating their duties. Indeed, Jewish and Islamic scriptures describe God as experiencing emotions of anger and hatred which move Him to punish the wicked (see, for example, *Isaiah* 42:25; *Proverbs* 6:16–19; *Quran* 40:10, 60:13). On the other hand, it is recognized that no human can perfectly obey the law. Jewish and Islamic scriptures also describe God as experiencing emotions of love and compassion, which move Him to forgive wrongdoing and grant humans blessings which they do not deserve (see for example, *Deuteronomy* 7:7, 13; *Isaiah* 55:7; *Hosea* 3:1; *Quran* 2:195, 5:39, 16:18, 55–1:78; Shah-Kazemi 2010).

In Christianity, matters are somewhat different. Christianity has a more complex attitude towards the domain of politics, law, and war. Christianity grew out of a (proto-) Jewish tradition, and assigned Jesus the status of Messiah. According to traditional Jewish understandings, the Messiah is a future mighty warrior king like David. The Messiah is expected to save the Jewish people by securing their political independence. Hence, he will use war to free them from foreign domination and then implement proper Jewish law. Christians believe that this traditional Jewish concept of the Messiah is partly mistaken and partly correct. It is mistaken in that it is overly concerned with politics, law, and war. However, it is correct in holding that the Messiah offers salvation to the Jews. For Christians, the Messiah not only offers salvation to the Jews, but to all of humankind. However, this salvation is not political in character. Rather, it is an otherworldly salvation, attained through faithful acceptance of Jesus as the divine Messiah, and proper moral behavior in the domain of individual ethics.

Christianity places special emphasis on love (*agape*), and a kindness-oriented moral code (see Schoenfeld 1989; Grant 1996; Jennings 1996; Jackson 2003; Wolterstorff 2015; Cochran and Calo 2017). In the *New Testament*, Jesus is depicted as teaching that morality centers on love of God and love of other humans (i.e., "love of neighbor") (*Matthew* 22:34–40, *Luke* 10:25–27; also see *Deuteronomy* 6:4–7). Indeed, Jesus goes so far as to reject the commonplace view that emotions of hatred and anger are valid when directed at enemies who cause one harm. Rather, Jesus exhorts his followers not to hate such people, but rather to love them (*Matthew* 5:43–48). In the *New Testament*, Jesus is repeatedly portrayed as advocating a moral standard higher than justice. He urges people not to demand punishment when they are assaulted ("turn the other cheek"), and to aid people who seek to rob them (*Matthew* 5:38–40). While being crucified, Jesus calls out in prayer to Father God, requesting that He forgive those carrying out the crucifixion (*Luke* 23:34). Jesus also works to undermine strict observance of various rules prescribed by Jewish law, including sabbath rules, dietary rules, and rules requiring stoning as a punishment for adultery (*Mark* 2:23–27, 7:1–23; *John* 7:53–8:11). Out of love, people are forgiven and spared from punishments that they deserve according to the law. Following Paul, Christians would come to hold that

Jesus abolished the general body of Jewish legal rules (i.e., the 613 *mitzvot*). In particular, Jesus rejected the ritual/symbolic rules which marked off and preserved the Jews as a unique people (e.g., circumcision, dietary laws, strict Sabbath observance).

Christian attitudes towards law and war are complicated. Generally speaking, premodern Christian thinkers do not reject the legitimacy of law and war per se. The *New Testament* depicts Jesus as arguing that love for others is the core principle of Jewish law (*Matthew* 22:34–40; *Luke* 10:25–27). Thus, in promoting love for others he is affirming the law. Moreover, although Christians posit that Jesus abolished the general body of Jewish legal rules, they assert that Jesus did not abolish the basic moral principles embodied in the Ten Commandments (e.g., prohibitions on murder, theft, and adultery) (Smith 2014).

Following Paul, early Christians held that God had granted authority to the non-Christian Roman emperors who ruled over them (*Romans* 13:1–7). As such, they obeyed Roman law. When the Roman emperor Constantine converted to Christianity, Christians ascended to political power. Subsequently, Christian rulers and Church authorities began endorsing legal codes based on Christian values—building on material from the *Old Testament* (e.g., the Ten Commandments), the *New Testament*, Church canons, and Roman law (Helmholz 1996; Witte 2002; Tuininga 2017; also see Uhalde 2007). In developing laws, Catholic, Orthodox, and Protestant Christians also relied on notions of natural law. As indicated earlier, notions of "natural law" are often tacitly tied to biologically-rooted moral intuitions. The Ten Commandments were widely seen as embodying natural law principles (Helmholz 2017). Notable advocates of natural law included Paul (esp. *Romans* 2:12–15), John Chrysostom, Augustine, and Thomas Aquinas (Doe 2017; Helmholz 2017). Aquinas is the foremost Christian expositor of natural law. He held that humans naturally incline towards a limited number of basic desires. These include the desire to preserve one's life and bodily integrity; the desire to procreate; the desire to care for and raise the children produced through procreation; the desire to live peacefully in a group; and the desire to acquire various types of knowledge—the highest being knowledge of God (see O'Connor 1967, esp. p. 62; Boyd 2004; Beckwith 2021). In the view of Aquinas, reason tells us that laws should be designed to help fulfill these basic human desires. For example, murder and assault should be banned in order to help preserve human life and bodily integrity. Contraception should be banned to help ensure that the species is continued through reproduction.

Building on earlier Greco-Roman ideas (Keller 2012; O'Driscoll 2015), Christianity also developed a just war theory to legitimate and regulate war (Corey and Charles 2012). Christian thinkers generally endorsed defensive war, and some additionally supported certain types of offensive war. In all cases, military violence was to be limited by moral considerations (Kolbaba 1998; Rivera 1992, pp. 235–57; Jenkins 2011; Corey and Charles 2012; Tellkamp 2020, pp. 199–251).

Unlike Judaism and Islam, Christianity has a strong monastic tradition (see Wimbush and Valantasis 1998; Innemee 2005; Olson 2008). Both males and females can be monastics. The *New Testament* presents Jesus as a monastic. He embraces celibacy and poverty. He cares not for his own body, willingly accepting martyrdom through gruesome torture and crucifixion. Christian monastics were bound by special ascetic norms inspired by Jesus. These norms prescribed celibacy and poverty while encouraging martyrdom (see Moss 2012). The norms also mandated an enhanced kindness-oriented moral code, which eschewed violence while prioritizing the domain of individual ethics. However, this did not prevent monastics from endorsing just wars. Indeed, the Middle Ages famously witnessed the foundation of Catholic monastic military orders, whose members took up arms and directly participated in the Crusades and other conflicts (e.g., Knights Templar, Knights Hospitaller, Teutonic Knights) (Barber 1994; Riley-Smith 2012). Still, the dominant forms of monasticism lacked a military component.

Following the time of Constantine, Church authorities customarily exerted significant power in the domain of politics (e.g., crowning rulers[7], excommunicating and deposing rulers[8], granting rulers control over particular lands and peoples[9]). Nevertheless, primary

responsibility for the political domain was assigned to lay Christian kings (supported by armies), who were charged with maintaining justice. The classic Christian theory of the "two swords" asserted that God had given power over otherworldly affairs to the Church (including its monastics and priests). God had given a complementary power over worldly affairs to lay Christian kings (Helmholz 1996, pp. 339–65).

Notably, in the premodern period, Christian thinkers criticized Islamic doctrine as excessively violent (e.g., teachings legitimating offensive war). Meanwhile, Muslim thinkers criticized Christian doctrine as impractical and socially destructive (e.g., the idea that a person should allow others to beat him with no consequences, or forego marriage and reproduction to be a virgin) (see, for example, Lazarus-Yafeh 1996, p. 73; Tolan 2002, pp. 94, 149, 249–54; Pearse 2009; also see Reinert 1991; Celik 2017). In such exchanges, both Christians and Muslims often interpreted each others' doctrines in an uncharitable manner. Christians downplayed the fact that violent Islamic policies were legitimated with reference to justice. They also overlooked similar policies within Christianity. Meanwhile, Muslims suggested that Christians intended the entire population to adopt the demanding regulations of monastics. Additionally, Muslims downplayed similar forms of voluntary asceticism within Islam (Tolan 2002, pp. 94, 149, 249–54; Pearse 2009; also see Reinert 1991). (Significantly, Muslims not only accepted Jesus as a Prophet, but frequently portrayed him as an ascetic Sufi Muslim mystic; see Lazarus-Yafeh 1996, pp. 82–83).

Christianity's moral prioritization of kindness over justice shapes its theology. Like Judaism and Islam, Christianity conceptualizes God as a just king who rewards and punishes people in accordance with their deeds, while also exhibiting some measure of forgiveness. Nevertheless, in Judaism and Islam, it is emphasized that obedience to the law makes one deserving—to some extent—of God's reward. By contrast, Christianity does not assert that legal obedience makes one deserving, and even deemphasizes deservingness based on morally good actions. (Such ideas are found in the writings of Paul and Augustine, but are taken furthest by Martin Luther) (Gundry 1985; Dunn and Suggate 1993; DeVries 2007; Karfikova 2012). Rather, Christianity emphasizes divine kindness in the form of undeserved forgiveness. Hence, ancient Judaism teaches that God is moved to forgive sin when He receives animal sacrifices at the Jerusalem temple. According to Christianity, God sacrificed His own son via crucifixion in Jerusalem like a "lamb", so that humanity's sins can be forgiven (see, for example, *John* 3:16; *I Corinthians* 5:7; *Ephesians* 1:7; *Hebrews* 9:22). Christians believe that human salvation, and heaven, are made possible through this act of divine forgiveness. God grants humankind forgiveness (as a type of grace) despite the fact that they are not deserving (Gundry 1985; Dunn and Suggate 1993; DeVries 2007; Karfikova 2012).

## 6. Section (V): Patterns in the Indic Religious Traditions

Biologically rooted moral intuitions also give rise to particular patterns in the Indic religious traditions. At the most general level, these patterns are reflected in attitudes towards kindness, peace, and love versus justice, violence, and hate. At a more specific level, these patterns are reflected in: (1) doctrines concerning politics, law, and war; (2) doctrines concerning individual ethics, and moral behavior proper to monastics and laypersons; and (3) doctrines concerning theological matters, such as the nature of the universe, souls, and deities. This section will explore the preceding patterns in some detail.

Like the Abrahamic traditions, the Indic traditions value a kindness-oriented moral code defined by kindness, peace, and love. They likewise value a justice-oriented moral code which legitimates violence and hatred under certain circumstances. However, there are differences in emphasis. Here, it is useful to distinguish between Hinduism on the one hand, and Buddhism and Jainism on the other. Hinduism places comparatively greater emphasis on the justice-oriented moral code, and, in this sense, resembles Judaism and Islam. Meanwhile, Buddhism and Jainism place comparatively greater emphasis on the kindness-oriented moral code, and, in this sense, resemble Christianity. The aforementioned differences in emphasis shape doctrines on politics, individual ethics, monasticism, and theology. Let us examine these issues in more depth.

Within Buddhism, Jainism, and a major current of Hinduism (i.e., Yoga), it is believed that morality centers on five general principles. These include: (1) refraining from violence; (2) refraining from stealing; (3) refraining from lying; (4) refraining from sexual immorality (e.g., adultery if married, any sex if one is a monastic). There are differences regarding the fifth principle. Buddhism holds it to be refraining from intoxicant consumption (similar to Islam). In Jainism and Hinduism, the fifth principle is refraining from greed and overattachment to worldly things (*Aparigraha*) (Harvey 2000, pp. 60–87; Keown 2005, pp. 8–10; Long 2009, pp. 101–10; Howard 2018).

The Indic traditions accord unique importance to the principle of refraining from violence, which is known as "*ahimsa*" (Harvey 2000, p. 69; Gier 2004, pp. 28–38, 51–65; Long 2009, pp. 99–115; Howard 2018; also see Alsdorf 2010). On the most basic level, *ahimsa* simply means not committing violence against others—especially physical violence (e.g., killing, striking). In this sense, it overlaps with standard norms against killing and assaulting humans found in the Abrahamic traditions (e.g., Ten Commandments, Theory of *Maqasid al-Sharia*). Nevertheless, in the Indic traditions, the principle of refraining from violence is taken further than in the Abrahamic traditions. In the Indic traditions, it is not only applied to humans, but also to animals and even plants. *Ahimsa* can be thought of as a commitment to peace, which is part of a broader commitment to kindness. Moreover, the type of kindness involved is markedly universal (rather than parochial) in character. Thus, one refrains from inflicting violence on various types of beings (human and non-human) as one wishes to show all of them kindness.

Hinduism, Buddhism, and Jainism have strong traditions of monasticism—in which men (and sometimes women) participate (Babb 1996; Flood 1996; Dundas 2002; Gombrich 2006; Long 2009; Strong 2015). Monastics have a special code which requires celibacy, poverty, and the wearing of minimal or no clothing. The special code also mandates a heightened commitment to *ahimsa*. Put differently, monastics must embrace an enhanced kindness-oriented moral code. In keeping with this code, monastics renounce direct participation in the domain of politics (e.g., enforcing laws, fighting wars). Understanding morality in the Indic traditions requires careful attention to the distinction between monastics and non-monastics.

Like Judaism and Islam, Hinduism accepts the domain of politics as perfectly legitimate, and holds that proper laws and wars are morally good. Hinduism prescribes a religious law (*dharma*), akin to Jewish law and Islamic law (Rocher 1978; Menski 2006, pp. 196–278; Davis 2010; Olivelle and Davis 2018). This Hindu law serves as a mechanism for preserving social relationships (e.g., parent-child relationships, marriage relationships, kin relationships).

Hindu law holds that different social groups should have different moral codes. More specifically, Hindu law divides people into four different castes (*varnas*), each with a specific set of duties (Davis 2010). There is the *Brahmin* caste of priests and monastics; the *Kshatriya* caste of kings and warriors; the *Vaishya* caste of farmers, pastoralists, and merchants; and the *Shudra* caste of servants and slaves. As mentioned earlier, Jewish law and Islamic law are concerned with preserving the Jews and Muslims as groups. It is debatable whether Hindu law seeks to preserve Hindus as one unified group. Nevertheless, it can be said that Hindu law seeks to preserve each of the castes as a group by giving it a distinctive way of law defined by particular rules (e.g., rules regulating labor, purity rules, rules for worship). Hindu law links each caste group with other groups in a hierarchy. *Brahmins* are on top, followed by *Kshatriyas*, followed by *Vaishyas*, followed by *Shudras*. All caste groups are supposed to cooperate together for the good of society by performing their designated roles.

Hindu law assigns *Brahmins* a moral code which centers on *ahimsa* (i.e., an enhanced kindness-oriented moral code). *Brahmins* should avoid directly participating in political violence (see Gier 2004, pp. 34–36; Bronkhorst 2016, pp. 72–73). Ideally, *Brahmins* embrace vegetarianism so as not to inflict violence and killing on animals (see Alsdorf 2010). Ide-

ally, *Brahmins* also become full monastics once they have already built a family and left descendants (i.e., in keeping with the *Varnashrama dharma* system; Olivelle 1993, pp. 131–60).

While Hindu law considers *ahimsa* admirable, it holds that absolute nonviolence is not morally appropriate for *Kshatriyas*—a view famously expounded in the *Bhagavad Gita* (Miller 1986; Sharma 2003; Gier 2004, pp. 34–38; Bronkhorst 2016, pp. 69–73). Rather, *Kshatriyas* must adopt a justice-oriented moral code. Hence, they have a duty to impose Hindu law and to fight wars, as these are necessary to protect society and its members (Flood 1996, pp. 71–72; Subedi 2003, esp. p. 345; Allen 2006). Hinduism endorses defensive war as well as offensive war, and the ideal Hindu king is a *Chakravartin* or "world-conquering ruler" (Bronkhorst 2016, pp. 69–73). War should be waged in keeping with moral principles of justice. Thus, Hindu law forbids the killing of women, children, the aged, and *Brahmin* monastics (Subedi 2003; Allen 2006; Dwivedi 2017a). Although *Kshatriyas* are responsible for establishing justice, they are (like all Hindus) still encouraged to exhibit the *ahimsa*-related values of kindness, peace, and love. *Kshatriyas* can implement these values to some extent in the domain of politics. Nevertheless, there is most room to implement them in the domain of individual ethics (e.g., in personal interactions not tied to government duties).

As noted earlier, monasticism is often associated with the notion that there are different moral codes for different groups which cooperate together in a society. Recall, for instance, that the Christian Church and its monastics have a higher kindness-oriented moral code, but they cooperate with lay Christian kings, and their armies, who have a lower justice-oriented moral code. Hinduism manifests a variant of this phenomenon. *Brahmins* and their kindness-oriented moral code are assigned a higher status. *Kshatriyas* and their justice-oriented moral code are assigned a lower status. Nevertheless, *Brahmins* cooperate with *Kshatriyas* for the good of society.

Like Christianity, Buddhism and Jainism have a more complex and ambivalent relationship towards the domain of politics, law, and war. Christianity emerged out of an earlier (proto-) Jewish tradition, but did not acknowledge the general body of Jewish law as binding. Similarly, Jainism and Buddhism emerged out of an earlier (proto-) Hindu tradition, but they did not acknowledge the general body of Hindu law as binding.

There are notable parallels between Christianity, Buddhism, and Jainism in how they understand their founding figures. Thus, Christianity was founded by Jesus, whereas Buddhism and Jainism were founded (respectively) by Siddhartha Gautama and Mahavira.[10] It is believed that Gautama and Mahavira were members of high-ranking *Kshatriya* royal families, and were in line to rule. Like Jesus, both left the domain of politics and became monastics devoted to saving humanity through their spiritual teachings. Buddhism and Jainism divide society into monastics and laypersons (endorsing a simpler view of society than that underlying the Hindu caste system). Just as Jesus is the model monastic for Christians, Gautama and Mahavira are the model monastics for Buddhism and Jainism.

Buddhism champions *ahimsa*, which it associates with universal kindness as well as emotions of love and compassion (*metta*, *karuna*) (Jennings 1996; Harvey 2000, pp. 103–9). Moreover, Buddhism condemns hatred and anger, as they are linked to violence (Harvey 2000, pp. 10, 17, 105; Keown 2005, pp. 36, 70). Commitment to *ahimsa* caused some Buddhist schools to take up vegetarianism, even if others did not (see, for example, Gier 2004, p. 52; Williams 2009).

Like Jesus, Gautama is depicted as championing love and compassion towards all people, and in all circumstances—even towards armed robbers who are in the process of sawing off one's body parts (Harvey 2000, p. 105). Gautama is held to have lived some 550 previous lives in the form of animals, gods, and humans (i.e., as recounted in the *Jataka* tales; see for example, Khoroche 1989; Cone and Gombrich 1977; Shaw 2006). In these previous lives, Gautama's love and compassion led to him to make various sacrifices as acts of kindness. Hence, he sometimes sacrificed his own life for others in a manner akin to Jesus (e.g., Khoroche 1989, pp. 213–20). Gautama's extraordinary kindness is epitomized in his past life as Prince Vessantara, when he embraced an attitude of unlimited charity. Whenever the Prince was asked for anything, he freely gave it away. He did so no matter

how outrageous the demand, and regardless of the demand's negative consequences for himself, his family, and the kingdom he ruled. As acts of charity, the Prince gave away his kingdom's source of food (exposing his subjects to death and famine). He also gave away his children as slaves, and then gave away his beloved wife to another man (see Cone and Gombrich 1977; Keown 2005, pp. 13–14). The tale of Prince Vessantara recognizes that unlimited kindness threatens society and all social relationships, but praises it nonetheless—although in the tale, negative consequences are ultimately averted through intervention of the gods.

Within Buddhism, the emphasis on love and compassion intensified over time. Earlier Theravada doctrine teaches that a person should embrace kindness, love, and compassion to achieve his/her own individual salvation. Salvation entails disappearing from the world into a state of *nirvana*. Later Mahayana doctrine teaches that a person should not simply focus on his/her own salvation. Rather, as an act of universal kindness, s/he should take a vow to help all beings attain salvation/*nirvana* (i.e., the *Bodhisattva* vow). An individual should fulfill the vow (over the course of many lives) before fully entering into *nirvana* himself/herself (for a discussion of this complex topic see Williams 2009, pp. 55–62). From the Mahayana standpoint, the Theravada position is inferior in that it is not sufficiently loving and compassionate. Dating back to the premodern period, Buddhists have condemned Hinduism, Islam, and other religions for their excessive violence (Truschke 2021, pp. 36–38).

The principle of *ahimsa* is given the widest application in Jainism (Sharma 2003, p. 503; Gier 2004, pp. 28–34; Long 2009, pp. 99–115). Hence, lay Jains reject killing animals and plants. Consequently, they do not eat animal meat. They also avoid killing plants by only eating parts of them, and not uprooting them. Jain monastics are held to an even higher standard. They wear mouth coverings to avoid accidentally ingesting and killing small organisms. They also carry brooms to sweep the ground in front of them such that they do not accidentally step on small organisms. At the same time, it is recognized that so long as one lives, one will inevitably kill some organisms, even if unintentionally. Thus, the ideal of Jain monks is to fast until death (*sallekhana*) (Long 2009, pp. 110–11). Jains condemn other religious traditions, such as Buddhism and Hinduism, for their excessive violence (Gier 2004, p. 52; Dwivedi 2017b).

Although Buddhism and Jainism hold that forsaking politics in favor of absolute nonviolence is the morally best option, they do not categorically reject politics, law, and war. Generally speaking, they tend to reluctantly accept such things as necessary, or even praiseworthy, in certain circumstances. Gautama's example provides a precedent for the notion that political rule can be morally good. It is held that in previous reincarnations, Gautama passed virtuous lives as a human ruler (e.g., Makhadeva, Mahajanaka, Vessantara) (see Bronkhorst 2016, pp. 78–82). He likewise passed virtuous lives as a god ruling over other gods (e.g., Indra), and an animal ruling over other animals (e.g., monkeys, birds) (see Khoroche 1989; Cone and Gombrich 1977; Shaw 2006). More generally, Buddhist texts praise kings who embrace Buddhism, rule in keeping with its teachings, and establish justice (e.g., Ashoka, Milinda). While ruling, the king himself is not a monastic. Nevertheless, he is supposed to cooperate with monastics (Friedlander 2009; Moore 2016).

There are many examples of premodern Buddhist legal systems which blend Buddhist norms with Hindu legal norms and/or local cultural norms (e.g., in Myanmar, Thailand, Tibet, Mongolia, Sri Lanka). Unsurprisingly, such legal systems make use of violent punishments to establish justice (see French 2002; French and Nathan 2014; Baker and Phongpaichit 2016; Bronkhorst 2016, pp. 77–82). Buddhist texts describe a king as someone who shows "anger where anger was due, censure those who deserved it, and banish those who deserved banishment" (Bronkhorst 2016, p. 74). Buddhist texts do not lay down a consistent set of regulations on warfare. Nevertheless, they frequently justify warfare (especially defensive warfare) and insist that it be constrained by moral principles (Jerryson and Juergensmeyer 2010; Bronkhorst 2016; Jenkins 2016; Sugiki 2020).

Many Buddhist texts hold that acts of violence associated with law and war have different aspects. Given their violent nature, such acts have a morally bad aspect, which generates negative karmic punishments. Nevertheless, if the acts operate to establish justice and protect people, they also have a morally good aspect, which generates positive karmic rewards. When an act is more good than bad, the karmic rewards can exceed and cancel out the karmic punishments. Consequently, a king can expect otherworldly rewards for using violence to establish justice and protect people (Jenkins 2016; Sugiki 2020; also see Jerryson 2016, p. 123–24). However, some Buddhist texts maintain that violence is problematic even for a king. Hence, a good king is best advised to simply abdicate and become a monastic (Bronkhorst 2016, p. 75–77). Other Buddhist texts seek to circumvent the problem of political violence through appeal to the supernatural. Hence, it is claimed that when a Buddhist king is truly righteous a supernatural wheel will appear in the sky. All who see it will be impressed, take it as a sign of the king's power, and submit to him. In this way, the king is not only able to govern without using violence, he can also conquer the world without violence and become a *Chakravartin* (see Bronkhorst 2016, pp. 74–75; Moore 2016, p. 19).

Jainism also grants some legitimacy to politics and political violence. Jain texts praise Jain kings, and non-Jain kings who treated Jains well (Cort 1998, pp. 85–111), while providing some justifications for war (Gier 2004, p. 29; Dwivedi 2017b). Like Gautama, Mahavira passed virtuous previous lives as a human ruler (e.g., Priyamitra, Nandana).

Although Buddhist and Jain kings are responsible for establishing justice, they are (like all Buddhists and Jains) still encouraged to exhibit the *ahimsa*-related values of kindness, peace, and love. Kings can implement these values to some extent in the domain of politics. Nevertheless, there is more scope to implement them in the domain of individual ethics (e.g., in personal interactions not tied to government duties).

It should be noted that while Indic monasticism sought some distance from the domain of politics, monastics often exerted significant influence over this domain—even if only indirectly. This is true in the premodern period, and remains true at present (e.g., monastic involvement in contemporary Buddhist and Hindu nationalisms; Pinch 1996; Banerjee 2005; Jerryson 2011). Indeed, the premodern period witnessed the emergence of warrior monks within Buddhism and (to lesser extent) Hinduism (e.g., *sohei* in Japan, *naga sadhus* in India) (Lorenzen 1978; Jerryson and Juergensmeyer 2010; also see Pinch 1996). Thus, like members of Catholic military orders, some Buddhist and Hindu monastics were willing to directly participate in warfare to advance political and religious aims. Nevertheless, it should be emphasized that warrior monasticism was never the typical form of monasticism characteristic of Christianity, Buddhism, or Hinduism.

The moral teachings of Hinduism, Buddhism, and Jainism shape their theological doctrines. Indic support for *ahimsa* (i.e., non-violence) towards both humans and non-humans is tied to the Indic doctrine that non-humans (e.g., animals, plants) have human-like souls. Thus, through reincarnation, any animal has likely already lived as a human in the past and will likely again become a human in the future (Long 2009, p. 182; Chapple 2017). By contrast, the Abrahamic religions are minimally concerned with violence towards non-humans because it is held that they lack human-like souls. Indic moral teachings also shape how the highest and most honored deities are portrayed. Thus, because Hinduism has a positive view of just violence, its gods directly participate in wars, and inflict violence on evildoers (e.g., in *Mahabharata* and *Ramayana*) (see Bronkhorst 2016, p. 88). Meanwhile, as noted above, Buddhism and Jainism place special emphasis on kindness, love, and peace. This is reflected in their portrayals of *Buddhas* and *Tirthakaras/Jinas*. These enlightened god-like beings teach others the path to salvation and are the highest moral exemplars. They are portrayed as calm, loving, and compassionate in nature. Unlike the Abrahamic God, they do not experience anger and hatred, which makes sense given that they do not punish individuals for evil deeds. Rather, evil deeds are punished by the law of *karma* (and lower gods).

## 7. Conclusions

In modern Western societies, it is common to identify kindness, peace, and love with moral goodness, while identifying violence and hatred with moral evil. These standards are used to render judgement on religion in general, and on specific religious traditions. However, cognitive science research on biologically rooted moral intuitions indicates that matters are far more complicated. Drawing on this research, the present article has explained that violence, hatred, and justice are interrelated psychological phenomena. Moreover, these phenomena play an essential role in preserving the valued cooperative social relationships upon which human survival and reproduction depend. At the same, these social relationships also depend on kindness, peace, and love. Consequently, every society—and every religion—must grapple with the challenge of balancing justice, violence, and hatred with kindness, peace, and love.

The article set forth a framework for addressing this issue. The article argued that biologically rooted moral intuitions give rise to two moral codes. There is a kindness-oriented code associated with the domain of individual ethics, as well as a justice-oriented code associated with the domain of politics. The Abrahamic and Indic traditions accept both codes, but balance them in different ways, giving rise to distinctive patterns. At the most general level, these patterns are reflected in attitudes towards kindness, peace, and love versus justice, violence, and hate. At a more specific level, these patterns are reflected in: (1) doctrines concerning politics, law, and war; (2) doctrines concerning individual ethics, and moral behavior proper to monastics and laypersons; and (3) doctrines concerning theological matters, such as the nature of the universe, souls, and deities. While numerous existing studies address the preceding attitudes and doctrines, they do not explain them in relationship to larger patterns produced by biologically rooted moral intuitions.

Clarifying these claims, the article argues that Judaism, Islam, and Hinduism adopt a similar approach to balancing the justice-oriented moral code and the kindness-oriented moral code. The three religious traditions have a positive view on the domain of politics, and place comparatively greater emphasis on the justice-oriented moral code associated with this domain. The article also argues that Christianity, Buddhism, and Jainism adopt a similar approach to balancing the moral codes. Hence, all have more complicated views on the domain of politics. They place comparatively greater emphasis on the kindness-oriented moral code associated with the domain of individual ethics. At the same time, the article also argues that apparent differences between the aforementioned religious traditions are not as pronounced as they initially appear. Judaism, Islam, and Hinduism all encourage kindness, peace, and love in the domain of individual ethics—even if they do not emphasize these values as much as Christianity, Buddhism, and Jainism. Similarly, while Christianity, Buddhism, and Jainism have more complex attitudes towards political violence (i.e., law, war), they generally acknowledge that it is necessary for establishing justice, and can even see it as morally good.

Although a distinction between Judaism/Islam/Hinduism and Christianity/Buddhism/Jainism is helpful in highlighting particular patterns, it is admittedly somewhat simplistic, and obscures various similarities and differences between the relevant traditions. For example, compared to Buddhism and Jainism, the Christian tradition gives more attention to politics, law, and war. In this sense, it is somewhat closer to Judaism, Islam, and Hinduism. Moreover, unlike Judaism and Islam, Hinduism gives significant attention to a kindness-oriented monastic code. In this sense, it is somewhat closer to Christianity, Buddhism, and Jainism.

That being said, identifying patterns generated by biologically rooted intuitions is still useful. The patterns provide a novel framework for analyzing the world's major religious traditions. More specifically, the patterns offer a new way of explaining and comparing their attitudes towards kindness, peace, and love as well as justice, violence, and hate. The patterns also offer a new way of explaining and comparing their various doctrines.

**Funding:** This research received no external funding.

**Acknowledgments:** I wish to thank the editors and reviewers for their valuable input. It greatly improved the article.

**Conflicts of Interest:** The author declares no conflict of interest.

## Notes

1.  Consider the Abrahamic traditions. Scholars frequently use the term "Abrahamic traditions" as a means of highlighting linkages between Judaism, Christianity, and Islam (Peters 2004; Silverstein et al. 2015; Stroumsa 2015; Cohen 2020). Thus, the three traditions share in a historical genealogy; they have continuously interacted with one another throughout history, and they endorse many similar ideas and practices (e.g., monotheism, respect for Abraham, prayer). Nevertheless, the term "Abrahamic" traditions/religions remains controversial (Hughes 2012). This is largely because it downplays historical change and internal diversity within Judaism, Christianity, and Islam. For example, during the premodern period, the Jewish tradition underwent several major stages of development (e.g., Ancient Israelite religion, Second Temple Judaism, Rabbinic Judaism). Moreover, at each stage, variant forms of the Jewish tradition existed (e.g., Pharisees, Sadducees, and Essenes in Second Temple Judaism). Similar things can be said about the premodern Christian and Islamic traditions (e.g., the Islamic tradition underwent changes between the formative and classical periods; the Islamic tradition has Sunni, Shi'i, and Ibadi variants).

2.  Here, regression analysis is important. In regression analysis, statistical data is used to establish correlations between various factors (i.e., independent variables) and a phenomenon (i.e., dependent variable). Although correlation is not causation, in appropriate cases, correlation can be used as evidence of causation (e.g., a factor closely correlated with a phenomenon might causally influence that phenomenon). Researchers recognize that there are numerous factors which might causally influence religious and moral beliefs or behavior in a particular country (e.g., per capita income, average length of citizens' education, percentage of the populace that lives in cities, fertility rate, a history of communism, a history of colonialism). Regression analysis makes it possible to quantitatively measure correlational relationships between factors and a phenomenon (e.g., every 1000 dollar increase in a country's per capita income is correlated with—and conceivably causes—a one percent decrease in belief in God).

3.  Considerations of space also preclude detailed in-depth exploration of relevant religious doctrines, their historical development, and associated primary sources texts. Rather, doctrines will be described in an accurate, but concise and simplified, fashion.

4.  Buddhist doctrine technically rejects the notion of a persisting soul (i.e., the doctrine of *anatta*). However, Buddhists affirm something like a persisting soul.

5.  Or five.

6.  In Islamic law, the term "*mustahabb*" is used to describe praiseworthy supererogatory acts.

7.  E.g., Charlemagne.

8.  E.g., Henry IV of the Holy Roman Empire, Elizabeth I of England.

9.  E.g., Papal bulls *Dum Diversas*, *Romanus Pontifex*, *Inter Caetera*.

10. I.e., in the present age.

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
