# Peer review of "Comparing Moralities in the Abrahamic and Indic Religions Using Cognitive Science: Kindness, Peace, and Love versus Justice, Violence, and Hate"

_religions, doi:10.3390/rel14020203_

Round 1

Reviewer 1 Report (Previous Reviewer 1)

My only additional suggestions are with the new section on p. 4. The author uses a lot of causal language, but should instead avoid causal language because regression is correlational. 

- "Studies also demonstrate that religious traditions are one major cause of these differences" should be changed to "one major predictor of these differences

- "In regression analyses, statistical data is used to calculate how various factors . . . causally influence" --regression is not able to determine cause. 

- watch phrases such as "exert significant influence over" "affecting them", "different effects", as these all imply causality. Phrases such as these can be replaced by phrases such as "has a relationship with" or "is associated with, etc.

Author Response

Below, I explain how I revised the manuscript in keeping with the reviewers’ comments. Hence, I reproduce the comments from the reviewers' in italics. I then respond to the comments point-by-point. Where appropriate, I cite to pages within the revised manuscript.

Reviewer #1 Comments

My only additional suggestions are with the new section on p. 4. The author uses a lot of causal language, but should instead avoid causal language because regression is correlational.

- "Studies also demonstrate that religious traditions are one major cause of these differences" should be changed to "one major predictor of these differences

- "In regression analyses, statistical data is used to calculate how various factors . . . causally influence" --regression is not able to determine cause.

- watch phrases such as "exert significant influence over" "affecting them", "different effects", as these all imply causality. Phrases such as these can be replaced by phrases such as "has a relationship with" or "is associated with, etc.

I thank Reviewer #1 for these comments. In keeping with the comments, I have revised the manuscript to clarify that, strictly speaking, regression analysis only proves correlation rather than causation (see p.5-6, with footnotes). That being said, although regression analysis does not definitively prove causation, in appropriate circumstances, it can serve as evidence of causation. Moreover, the studies that I have cited use regression analysis as evidence of causation. In other words, such studies argue that religious traditions causally influence social behavior in statistically measurable ways. The revised manuscript indicates that relevant studies argue for causal influence (rather than mere correlation) (see p.5-6, with footnotes).   

Reviewer 2 Report (New Reviewer)

The article is, overall, attractive, well written and intellectually stimulating.

Nevertheless, I find two kinds of general problems in it.

1. Theological problems.

1.1 The author does not deal with primary sources, only secondary literature.

1.2 Probably because of the same reason, "Judaism" or "Islam" are reduced to signifiers without clear references. Judaism and Islam mean and can mean many different things. 

1.3 The use of the category of "Abrahamic" without any further comments is confusing.

2. Philosophical problem. 

The author does not define what is meant by the main concepts of the paper, "kindness", "peace", and "love", and it is not clear if they should be understood from a philosophical or social point of view.

I believe the article should still be published, but if these problems are not addressed, they should at least be briefly mentioned.

Author Response

Below, I explain how I revised the manuscript in keeping with the reviewers’ comments. Hence, I reproduce the comments from the three reviewers in italics. I then respond to the comments point-by-point. Where appropriate, I cite to pages within the revised manuscript.

Reviewer #2 Comments

The article is, overall, attractive, well written and intellectually stimulating.

Nevertheless, I find two kinds of general problems in it.

  1. Theological problems.

1.1 The author does not deal with primary sources, only secondary literature.

1.2 Probably because of the same reason, "Judaism" or "Islam" are reduced to signifiers without clear references. Judaism and Islam mean and can mean many different things.

1.3 The use of the category of "Abrahamic" without any further comments is confusing.

  1. Philosophical problem.

The author does not define what is meant by the main concepts of the paper, "kindness", "peace", and "love", and it is not clear if they should be understood from a philosophical or social point of view.

I believe the article should still be published, but if these problems are not addressed, they should at least be briefly mentioned.

I thank Reviewer #2 for his/her insightful comments, and kind words. I have revised the manuscript in keeping with these comments.

Thus, I explain why the manuscript “does not deal with primary sources, only secondary literature” (see p.6, with footnote).

I acknowledge that “Judaism and Islam mean and can mean many different things”. In the revised manuscript, I explain that these traditions have changed over the course of history, and exist in variant forms. Moreover, so matters are clearer for the reader, I specific that the manuscript focuses on “Rabbinic Judaism” and “classical Sunni Islam” (see p.4, 6, with footnotes).

In the revised manuscript, I discuss the concept of “Abrahamic” religious traditions and why some scholars find it problematic. I also include many citations to recent scholarship which utilizes the concept of Abrahamic traditions (see p. 4, with footnotes).  

Finally, in the revised manuscript, I clarify that the key concepts (e.g., kindness, peace, violence) used in the paper should be understood primarily from the standpoint of cognitive science research (rather than philosophy or social theory). Thus, in cognitive science, the term “kindness” is often used interchangeably with the terms “altruism” and “prosociality”. Moreover, the term “violence” is often used interchangeably with the term “aggression”. Meanwhile, “peace” is described as the absence of “aggression”. In the revised manuscript, I cite to relevant cognitive science studies which use these terms (see p.10-11).

That beings said, in the manuscript itself, I usually do not use technical cognitive science terminology (e.g., “prosociality”, “aggression”). Such terminology is often unfamiliar or confusing for scholars in the humanities, and I would like to make the manuscript accessible to a broad audience which includes scholars in the humanities.

Reviewer 3 Report (New Reviewer)

This is an important essay and most definitely deserves to be published as is.  It is an ambitious undertaking, and contributes to the ongoing interdisciplinary dialogue between the cognitive sciences and religious studies.  I share the author’s discomfort with the need to oversimplify the complexity of major world religions in an essay like this, but it is appropriate given the forward-looking nature of this piece.  It is well-researched, but some standard works are overlooked; for example, the very useful Moral Politics (1996), of cognitive linguist George Lakoff, which discusses the cognitive mental frameworks between “liberal” and “conservative” worldviews.  Although Lakoff addresses a specifically American context, his ideas might buttress the author’s arguments about the 2 models of “kindness” versus “justice.”

Because of the broad scope of the essay, some of the more specific findings of the cognitive science research seem to get lost in the narrative, which is also broken up by very long parenthetical reference citations.  In the well intentioned desire to be comprehensive, a certain methodological punch is lost by the selective and sweeping generalizations regarding Judaism, Christianity, Islam, Hinduism, Buddhism, and Jainism.  One cannot be an expert in all of these areas, but the discussions, especially of the Indic religions, tends to be a bit thin.  Again, given the brevity of such a journal article, this was perhaps unavoidable.  Moving forward, the author might consider a more granular analysis of specific historically-grounded examples to engage with the emerging cognitive science methodologies.  The essay is fine as it is, and it is solid step in the right direction.

As far as the discussion of monastics being largely aloof from worldly politics and in favor of the kindness model, in modern-day India we find any number of right wing “Hindutva” swamis involved in politics and in organized violence against Muslims and marginalized communities.  This is certainly at variance from most pre-modern monastic behavior and suggests a need to tweak the handling of monasticism as far as the two models.

In conclusion, this essay illustrates both the promise and the proverbial pitfalls of making bold interdisciplinary engagements between cognitive science and religious studies.  This often leads to disgruntlement from individuals in both areas of study.  On the one hand, one does have to oversimplify treatments of both areas; but such necessary reduction then leads to perceptions of thinness and under-consideration.  It is a classic dilemma, but it is a noble situation that is inevitable and part of the process of growing the disciplinary and interdisciplinary fields.  I congratulate the author for leading the way in this essay.

Author Response

Below, I explain how I revised the manuscript in keeping with the reviewers’ comments. Hence, I reproduce the comments from the three reviewers in italics. I then respond to the comments point-by-point. Where appropriate, I cite to pages within the revised manuscript.

Reviewer #3 Comments

This is an important essay and most definitely deserves to be published as is.  It is an ambitious undertaking, and contributes to the ongoing interdisciplinary dialogue between the cognitive sciences and religious studies.  I share the author’s discomfort with the need to oversimplify the complexity of major world religions in an essay like this, but it is appropriate given the forward-looking nature of this piece.  It is well-researched, but some standard works are overlooked; for example, the very useful Moral Politics (1996), of cognitive linguist George Lakoff, which discusses the cognitive mental frameworks between “liberal” and “conservative” worldviews.  Although Lakoff addresses a specifically American context, his ideas might buttress the author’s arguments about the 2 models of “kindness” versus “justice.”

I thank Reviewer #3 for his/her insightful comments, and kind words. I have revised the manuscript in keeping with these comments. The Lakoff reference is helpful, and I have integrated it into the revised manuscript (see p. 16)

Because of the broad scope of the essay, some of the more specific findings of the cognitive science research seem to get lost in the narrative, which is also broken up by very long parenthetical reference citations.  In the well intentioned desire to be comprehensive, a certain methodological punch is lost by the selective and sweeping generalizations regarding Judaism, Christianity, Islam, Hinduism, Buddhism, and Jainism.  One cannot be an expert in all of these areas, but the discussions, especially of the Indic religions, tends to be a bit thin.  Again, given the brevity of such a journal article, this was perhaps unavoidable.  Moving forward, the author might consider a more granular analysis of specific historically-grounded examples to engage with the emerging cognitive science methodologies.  The essay is fine as it is, and it is solid step in the right direction.

I appreciate the reviewer’s remarks, and acknowledge that the analysis is a “bit thin” in certain areas. That being said, the reviewer correctly recognizes that such “thinness” is “unavoidable” “given the brevity of such a journal article”.

As far as the discussion of monastics being largely aloof from worldly politics and in favor of the kindness model, in modern-day India we find any number of right wing “Hindutva” swamis involved in politics and in organized violence against Muslims and marginalized communities.  This is certainly at variance from most pre-modern monastic behavior and suggests a need to tweak the handling of monasticism as far as the two models.

In the revised manuscript, I have tweaked my analysis of monasticism. Hence, I am clearer that some monastics did exert influence in the political realm, and that some even took up the role of “warrior monks”. I also acknowledge that monastics play a role in contemporary Hindu and Buddhist nationalisms. I include appropriate citations for my analysis of these issues (see p. 22, 29).

In conclusion, this essay illustrates both the promise and the proverbial pitfalls of making bold interdisciplinary engagements between cognitive science and religious studies.  This often leads to disgruntlement from individuals in both areas of study.  On the one hand, one does have to oversimplify treatments of both areas; but such necessary reduction then leads to perceptions of thinness and under-consideration.  It is a classic dilemma, but it is a noble situation that is inevitable and part of the process of growing the disciplinary and interdisciplinary fields.  I congratulate the author for leading the way in this essay.

Thanks again to the reviewer for his/her kind words and encouragement.

This manuscript is a resubmission of an earlier submission. The following is a list of the peer review reports and author responses from that submission.

Round 1

Reviewer 1 Report

I found this paper fascinating. I especially liked how the author(s) discussed how both universal kindness as well as violence were linked to morality. In my field of social psychology, there is less linkage between certain conceptualizations of "violence", such as prejudice, and morality, but as the author notes, certain moral foundations such as Haidt's foundation of loyalty can be related to parochial kindness and even aggression/prejudice. I appreciated the link of both to morality. 

I also found it fascinating how the author noted that certain religions (Islam, Judaism, Hinduism) were related more strongly to justice, whereas others (Christianity, Buddhism) were related more to universal kindness. As it stands, the paper currently is divided into a section discussing Abrahamic religions, and then Indic religions. Given that some of the religions emphasize justice, and others universal kindness, it might help at the end of the paper to provide this new framework, summarizing the similarities that the Abrahamic and Indic justice religions have with each other, as well as the similarities that the Abrahamic and Indic universal kindness religions have with each others. The author does this a bit at the end of the Indic morality section, but this could be a little more comprehensive. 

It would also improve the paper if there was more of a conclusion at the end, rather than just a summary. What new information can we take from this synthesis and application? How should this new information make us look at these religions, and morality, differently? What new research questions can we glean from the application of cognitive science of religion to morality? 

Relatedly, it would be helpful if the author could more clearly note the contributions that this paper makes. What new information are we getting from the application of CSR to morality? What new things do we learn about these different religions, and how should this guide future research? The author might have already answered these questions, but as a reader outside of the cognitive science of religion literature, it was difficult for me to discern which information was part of the previous literature, and which information was a new contribution by this paper. Increased clarity on the original points and contributions made would be helpful. 

Lastly, there were several paragraphs that did not contain citations. As a social psychologist, I am used to seeing more citations for information. For example, when explaining the basics of different religious practices, I was expecting some citations but did not see any. 

In summary, this was an intriguing read. I'm interested to see some more compelling conclusions based on the interesting points and applications the author makes.

Author Response

In the email I received from the editor, I was asked to write a cover letter where I respond to comments from the editor and comments from the two external reviewers. 

I have written out a detailed 8 page cover letter which is attached. 

Reviewer 2 Report

The article provides a good overview of research on moral intuitions in cognitive science, and opens up an interesting and valid question how these intuitions are also relevant for understanding religious communities and teachings. However, the article tries to do too much for an article space: to compare Abrahamic and Indic traditions and how those teaching impact various large themes. In so doing, it creates very shallow and random stereotypes of these religious traditions and incorrectly contrasts Judaism&Islam and Christianity with each other, as if Christianity would be the major and unified tradition for universal kindness and monasticism. The presented data is too simplified. For this reason, the article should not be published as it is. The author should rather focus on some concrete cases, documents or time periods and do their comparison in a more rigorous way.

Author Response

(The authors gave the same response as above.)

Round 2

Reviewer 1 Report

I was glad to see the additional citations. I am used to seeing the citations presented in alphabetical order by first author, but perhaps other fields have different formats.